# Prediction of Protein Ion–Ligand Binding Sites with ELECTRA

**DOI:** 10.3390/molecules28196793

**Published:** 2023-09-25

**Authors:** Clement Essien, Lei Jiang, Duolin Wang, Dong Xu

**Affiliations:** Department of Electrical Engineering and Computer Science, Bond Life Sciences Center, University of Missouri, Columbia, MO 65211, USA; u.c.essien@mail.missouri.edu (C.E.); leijiang@missouri.edu (L.J.); wangdu@missouri.edu (D.W.)

**Keywords:** deep learning, ELECTRA, ion-binding site prediction, transformer, natural language processing, sequence-based prediction

## Abstract

Interactions between proteins and ions are essential for various biological functions like structural stability, metabolism, and signal transport. Given that more than half of all proteins bind to ions, it is becoming crucial to identify ion-binding sites. The accurate identification of protein–ion binding sites helps us to understand proteins’ biological functions and plays a significant role in drug discovery. While several computational approaches have been proposed, this remains a challenging problem due to the small size and high versatility of metals and acid radicals. In this study, we propose IonPred, a sequence-based approach that employs ELECTRA (Efficiently Learning an Encoder that Classifies Token Replacements Accurately) to predict ion-binding sites using only raw protein sequences. We successfully fine-tuned our pretrained model to predict the binding sites for nine metal ions (Zn^2+^, Cu^2+^, Fe^2+^, Fe^3+^, Ca^2+^, Mg^2+^, Mn^2+^, Na^+^, and K^+^) and four acid radical ion ligands (CO_3_^2−^, SO_4_^2−^, PO_4_^3−^, NO_2_^−^). IonPred surpassed six current state-of-the-art tools by over 44.65% and 28.46%, respectively, in the *F*1 score and MCC when compared on an independent test dataset. Our method is more computationally efficient than existing tools, producing prediction results for a hundred sequences for a specific ion in under ten minutes.

## 1. Introduction

Many biological processes are facilitated by the interactions between proteins and ligand ions [1]. These interactions are necessary for the proteins to carry out their functions properly [2,3]. More than fifty percent of proteins, when observed, interact with metal ions (cations) and acid radicals to stabilize their structure, and regulate their biological functions [4,5]. Fe^3+^ binding to hemoglobin is critical for transporting oxygen through the blood [6]. Ca^2+^ intracellular signaling triggers T-cell activation, and the development of B-cell response to antigens, differentiation, and development [7,8]. Zn^2+^ maintains the stability of the protein’s tertiary structure and is also essential for over 300 enzyme activities [9]—a lack or excess of it may cause central nervous system diseases [10]. The interaction of proteins with phosphate ions (PO_4_^3−^) can result in phosphorylation, which switches enzymes on and off, thereby altering their function and activity [11]. Sulfate ions (SO_4_^2−^) play a variety of structural roles, as well as binding to a variety of cytokines, growth factors, cell-surface receptors, adhesion molecules, enzymes, and fibrillary glycoproteins, to carry out various essential biological functions [12]. From these examples, we see that ions play significant roles in a wide range of cellular processes. Hence, the accurate identification of the protein–ion binding sites is important for understanding the mechanism of protein function and new drug discovery.

To understand the mechanism of protein–ion interactions, biological experiments, such as Nuclear Magnetic Resonance (NMR) spectroscopy [13] and fluorescence [14] methods, are carried out to measure the structure information of protein–ligand complexes and target ligand-binding proteins and their corresponding binding sites. As this is a very tedious and time-consuming process, computational methods to identify protein–ion binding sites are essential. The various computational methods proposed for predicting protein–ion binding sites can be grouped into sequence-based [15,16] and structure-based methods [17,18]. IonCom [19] proposed a new ligand-specific approach to predict the binding sites of nine metal ions (Zn^2+^, Cu^2+^, Fe^2+^, Fe^3+^, Ca^2+^, Mg^2+^, Mn^2+^, Na^+^ and K^+^) and four acid radical ion ligands (CO^2−^, PO^3−^, NO^2−^) using a sequence-based ab initio model that was first trained on sequence profiles, then extended using a modified AdaBoost algorithm to balance binding and non-binding residue samples. Sobolev and Edelman predicted the binding sites of protein chains and transition-metal ions by implementing the ‘CHED’ algorithm, obtaining a specificity of 96%; when predicting 349 whole proteins, 95% specificity was obtained [20]. Lu et al. used the fragment transformation method to predict metal ions’ (Ca^2+^, Mg^2+^, Cu^2+^, Fe^2+^, Mn^2+^, Zn^2+^) ligand binding sites, and obtained an overall accuracy of 94.6% and a sensitivity of 60.5% [21]. Ref. [22] identified four metal ions in the BioLip [23] database by implementing both sequence-based and template-based methods and obtained a Matthew’s correlation coefficient (MCC) greater than 0.5. Cao et al. used the SVM algorithm to identify ten metal ion-binding sites based on amino acid sequences, which obtained a good result using five-fold cross-validation [24]. Greenside et al. used an interpretable confidence-rated boosting algorithm to predict protein–ligand interactions with high accuracy from ligand chemical substructures and protein 1D sequence motifs, which produced decent results [25].

A major drawback of some of the existing computational tools is that they involve complex 3D computations, the threading of protein sequences to potential structural templates, and integrating multiple data types from both sequences and structures that are computationally intensive and time-consuming. In addition, several sequence-based tools have limited predictive performances (i.e., low precision) since they do not include tertiary structure information. 

In this work, we propose IonPred, a Deep Learning framework based on ELECTRA [26] for predicting ion binding in proteins. The model adopts a sequence-based method for predicting the binding sites of nine metal ions and four acidic radicals. It takes raw protein sequences from all the protein chains, with at least one binding site as input for the model. IonPred is based on the Transformer architecture, which adopts a two-stage pretraining and fine-tuning process. In the initial pretraining phase, it employs the replacement token detection technique to learn contextual representations within the protein sequences from unlabeled protein sequence fragments. In contrast, in the fine-tuning phase, the model is trained with labeled sequence fragments to perform various binary classification tasks for various types of ion-binding sites.

## 2. Results

To benchmark the performance of our method, we compared its predictive performance with existing tools, and selected the Zinc dataset as a case study to understand how its performance is affected by different model configurations.

### 2.1. Comparison with Other Tools

We compared IonPred with six state-of-the-art tools. Three of these are sequence-based tools (i.e., TargetS [27], ZinCaps [28], and LMetalSite [29]), while the other three are structure-based (i.e., MIB [30], IonCom, and DELIA [31]) predictors. ZinCaps only supports the prediction of Zn^2+^, while DELIA does not. We also compared the performance of our tool for predicting the binding sites of non-metal ions with IonCom. For the metals, as reported in Table 1, alkali metals (Na^+^ and K^+^) are the hardest to differentiate according to their low-performance scores, followed by the alkali earth metals (Ca^2+^, Mg^2+^). This could probably be due to the wide variability of ion binding in these ion categories, even among the homologous proteins, and subtle differences in their binding affinities across various amino acid residues. 

Except for Zn^2+^, where LMetalSite surpassed IonPred by 1.8% and 26.67% in the *F*1 and MCC, respectively, IonPred significantly outperformed all the sequence- and structure-based tools in most of the ion categories. The performance of LMetalSite is comparable to that of IonPred as both tools are based on a similar architecture (i.e., pretrained language models), and it shows that the sequence representation and contextual information learned from directly fine-tuning pretrained language models is both more insightful and more robust than when it is just used for feature extraction.

For the non-metal ion category, as seen in Table 2, IonPred outperforms IonCom in all metrics for all the acid radical by 50–117% for recall, 8.03–38.07% for precision, 44.65–67.03% in the *F*1 score, and 28.46–67.13% in the MCC.

We also plotted the ROC curves for the metal ions to further illustrate the superior performance of our method. As seen in Figure 1, except for Zn^2+^, the ROC curves for IonPred are all located at the upper portion of the plots to show more coverage and a higher AUC score. This indicates that IonPred has a greater capability to distinguish between positive and negative classes.

IonPred is far more computationally efficient than other tools, as it takes about 5 min to generate prediction results for 50–100 protein sequences of various lengths. It takes about 8 min to predict the same number of sequences with ZinCaps. It takes about 3 min to obtain the prediction results on just one protein sequence with TargetS, whereas it takes several hours to obtain one prediction result on just one protein sequence using IonCom and MIB.

### 2.2. Ablation Tests

To understand the efficiency of IonPred with different configurations, we used the Zn^2+^ dataset as a case study. This is because of its abundance in nature and the availability of quality datasets available for this ion. We evaluated the effect of the number of pretraining steps and the discriminator size on model performance. We pretrained and fine-tuned several ELECTRA models with various configurations for generator discriminator sizes, namely ELECTRA-0.25G-100K (the generator is 25% the size of the discriminator, with 100,000 training steps), ELECTRA-0.25G-200K (the generator is 25% the size of the discriminator, with 200,000 training steps), IonPred-0.25G-1M (the generator is 25% the size of the discriminator, with 1 million pretraining steps), ELECTRA-0.5G-200K (the generator is 50% the size of the discriminator, with 100,000 training steps) and ELECTRA-no-pretraining. We report their performance on the test dataset for Zinc in Table 3.

From the results, we see that of all the three models created with 200 K training steps, ELECTRA-0.25G-200K had the highest performance. This indicates that a generator size of 25% gives an optimal performance. Then, for all the configurations with a generator size of 0.25, we see that IonPred-0.25G-1M provided a better and overall superior performance. This indicates that a higher number of training steps gives a better performance. While ELECTRA-0.25G-100K has the same generator size as IonPred-0.25G-1M, it reports lower metric scores due to a lack of convergence of the model during pretraining. The model ELECTRA-no-pretraining, which was created without pretraining, reported the lowest performance for both AUC and AUPR.

### 2.3. Running Some Test Examples

We demonstrated the capability of our tool to identify ion-binding residues by running predictions on two known proteins that bind to Fe^3+^ and Mg^2+^, respectively. These proteins were obtained from RSCB. Table 4 contains the metal binding residues obtained from the protein database for each of the proteins, the position of the residue in the protein sequence, and the probability score from IonPred.

We see from the results that IonPred clearly identified three out of the five ion-binding residues in 3GKR_A for Fe^3+^ and accurately identified all six binding residues in 3DHG_D for Mg^2+^. Our model generally does a better job distinguishing residues that bind to Fe^3+^ than those that bind to Mg^2+^.

### 2.4. Tool

The pretrained ELECTRA model for ion-binding site prediction is provided as an open-source command-line tool, available at https://github.com/clemEssien/IonPred, accessed on 12 September 2023. It takes a Fasta file containing one or more protein sequences. Its instructions for use have been properly documented and the test datasets used are made available in the code repository. The output of the tool is a text file that contains the probability scores for each candidate site of the specified ion. The residues whose probability scores are higher than 0.5 are considered binding sites. IonPred was trained on a GPU, and it requires a GPU to run the prediction. The development environment requirements are Python 3, TensorFlow-GPU 1.15. CUDA 10, NumPy, Pandas, Scikit-learn, and SciPy. The default batch size for running predictions is 128.

## 3. Discussion

In this work, we presented IonPred, a pretrained ELECTRA model for predicting some of the most frequently seen ion-binding sites that have significant impact on protein structures and functions. Our method used raw sequence-based prediction because many proteins have no known structures or reliably predicted structures. The model was pretrained on a large corpus of unlabeled protein fragments in an unsupervised method and fine-tuned on a smaller quantity of non-redundant semi-manually curated labeled datasets. The model provided better predictive performance on alkali and alkali earth metal ions, which are typically difficult to predict. This is because the self-attention mechanism is adept at understanding the structural contexts of amino acid residues within protein sequences. This mechanism excels at assimilating conserved protein information by inherently focusing on neighboring residues and utilizes the transformer architecture to discern long-range dependencies.

However, there’s room for improvement for both metal categories. The attention mechanism of IonPred learns from the imbalanced dataset and provides improvement in the recall. We compared the different ELECTRA configurations of the training steps and generator sizes before we settled on the best configuration. IonPred significantly outperformed existing sequence and structure-based tools in all ion categories except Zinc, where LMetalSite slightly outperformed it. Here, we saw that directly fine-tuning the pretrained model on each specific binding site gave a better performance than just using it for feature extraction, as was demonstrated in LMetalSite.

The performance of the fine-tuning stage is mainly dependent on the availability of large high-quality labeled datasets. For ion-binding sites that have limited labels, its performance would not be as good. For future work, meta-learning could be explored as this could speed up the adaptation of binding sites with very limited labels. Also, the use of large protein information like ESM [32] or Sequence profile, and predicted structures from alpha fold [33] could also be incorporated to improve context-dependent biological properties learned by the discriminator with the purpose of significantly improving the recall.

## 4. Materials and Methods

### 4.1. Data and Data Processing

This study developed a new pretraining dataset by first downloading all protein chains from RCSB Protein Data Bank [34] using Biopython [35]. A total of 521,419 chains with their corresponding protein sequences were obtained. We excluded RNA and DNA components and protein chains that had less than 50 amino acid residues. We then made a series of API calls to the RCSB graph-based API [36], passing each protein chain ID and the keyword ‘UNIPROT’ as the parameters. The API response contained a lot of information, but we were only interested in the annotations. We obtained a total of 27,626 unique ligand-binding sites. While we identified various categories of binding sites, such as anions, cations, organic compounds, etc., we only focused on anions and cations for this study. Then, we used the sliding-window technique to extract fragments of a length of 25 (i.e., 12 amino acid residues on each side of the candidate binding residue). We chose fragment lengths of 25, because from the literature, fragment lengths of 7–25 have been tested, and it has been demonstrated by different methods that optimal fragments vary between 17 and 25. We used positive fragments for pretraining because through experimentation, we determined that pretraining with positive fragments made it easier to learn features related to ion-binding fragments more effectively. This process is illustrated in Figure 2.

### 4.2. Candidate Residue Selection

Almost all the amino acid residues are potential binding sites to varying degrees. A few of them participate more frequently in ion binding than others. Some of these residues are regular candidates for specific ions. To determine which candidate binding residues should be the focus when applying the sliding window for fragment extraction (which was used to extract positive fragments, as illustrated in Figure 2), we used a binary heatmap to plot the distribution of each amino acid residue with respect to the number of ion-binding sites. Through this process, we plotted two sets of heatmaps for the thirteen ions from the IonCom dataset (Figure 3a,b) and from BioLip database (Figure 3c,d). The *x*-axis represents the twenty amino acids, while the *y*-axis represents the ion ligands.

The plot is a frequency distribution of amino acid residues in relation to the number of ion-binding sites. From the figure, we determined that a residue was a binding site if its frequency was greater than or equal to the mean of the total residues in each row for a particular binding site. For the metal ions in Figure 3a, every amino acid residue was a candidate residue, but we observed the highest representation of candidate residues at Aspartate (D), Glutamate (E), and Histidine (H), followed by Leucine (L) and Cysteine (C). For the acid radical ions, we observed a high frequency of candidate residues at Histidine (H), Arginine (R), Glycine (G), Threonine (T), Lysine (K), and Serine (S), and using the sliding window technique, we extracted protein fragments of a length of 25 (i.e., 12 amino acid residues to the left and right of each candidate residue of interest), as implemented in [30], around the following amino acid residues: CYS (C), ASP (D), GLU (E), GLY (G), HIS (H), THR(T), LYS (K), ARG (R) and SER (S) at the center. If the amino acid residue at the center was an ion-binding site, the whole fragment was considered a positive sample; otherwise, it was regarded as a negative sample. We excluded negative fragments that contained a binding residue.

After eliminating duplicate fragments and excluding the negative fragments, we obtained 283,526 positive fragments. The complete process for obtaining the protein sequences used, annotation, and input fragment generation for pretraining is summarized in Figure 2. For the second stage, where we needed labeled data for finetuning, we obtained the labeled data of nine metal ions and four acid radicals from IonCom. The CD-HIT [37] tool was used to split the fine-tuning dataset into training, test and validation sets using a 40% similarity threshold to avoid over-fitting. The distribution of proteins used for fine-tuning is displayed in Table 5.

We extracted fragments from the labeled fine-tuning dataset. To label the fragments, we used the candidate binding residues, determined from the distribution in Figure 3. Fragments extracted around a binding residue were labeled as positive examples, while fragments extracted around a candidate binding site that were not binding residues were labeled as negative training examples. The statistics of the training, test, and validation fragments are summarized in Table 6.

### 4.3. Problem Definition

The ion-binding-site prediction in this study was formulated as a binary classification problem. For example, given a protein sequence for which the binding sites are unknown, we selected a particular ion (i.e., Zn^2+^, Cu^2+^, Fe^2+^, Fe^3+^, etc.) for which we wanted to determine the binding sites. Then, the aim would be to ascertain if the candidate binding residues (from Figure 3) for the selected ion(s) were binding site(s) or not. This would output probabilities for each candidate residue. A probability of 0.5 and above was considered a positive prediction (i.e., an ion-binding site), while a probability less than 0.5 was regarded as a negative prediction (i.e., not a binding site).

### 4.4. Deep Learning Model

The architecture of the proposed IonPred, as shown in Figure 4, was based on the ELECTRA (i.e., “Efficiently Learning an Encoder that Classifies Token Replacements Accurately”) learning model. This architecture comprised two neural networks, a generator, and a discriminator. These networks basically mapped a sequence of input tokens $x=\left[x_{1},\ldots ,x_{n}\right]$ into a sequence of contextualized vector representations $h\left(x\right)=\left[h_{1},\ldots ,h_{n}\right]$.

So, for any given position *t*, where *x_t_* is a masked amino acid residue [MASK], the generator used a SoftMax layer to produce the probability of generating a particular token *x_t_*.
(1)$$pG\left(x_{t}|x\right)=\exp\left(e{\left(x_{t}\right)}^{T}h_{G}{\left(x\right)}_{t}\right)/\sum_{x^{\prime }}\exp\;(e{\left(x^{\prime }\right)}^{T}hG{\left(x\right)}_{t})$$

In the equation above, e denotes the embeddings for the amino acid residues. The generator was trained using masked language modeling (MLM). For a given input, $x=[x_{1},\ldots ,x_{n}]$, MLM selected a random set of positions ranging from 1 to *n* to mask out. This produced the vector $m=[m_{1},\ldots ,m_{k}]$. The residues in these positions were replaced with a [*MASK*] token, which was represented as $x^{masked}=REPLACE\;(x,m,\;[MASK])$. The generator learned to predict the original amino acid residues. The discriminator predicted whether the amino acid residue was originally from the input data or if it was a replacement from the generator distribution using a sigmoid output layer, as shown in the equation below:(2)$$D\left(x,t\right)=\mathrm{sigmoid}\left(w^{T}h_{D}{\left(x\right)}_{t}\right)$$

The masked-out residues were replaced by samples from the generator. This sample is represented as $x^{corrupt}$. The discriminator was trained to predict which residues in $x^{corrupt}$ matched the original input  x. The model inputs were described as shown below:(3)$$m_{i}\;\sim \;unif\{1,n\}\;for\;i=1\;to\;k\;\;\;\;\;\;\;\;\;\;\;\;{\;x}^{masked}=REPLACE(x,m,\;\left[MASK\right])$$
(4)$${\hat{x}}_{i}\sim p_{G}\left(x_{i}|x^{masked}\right\}\;for\;i\;\in mx^{corrupt}=REPLACE(x,m,\;\hat{x})$$

And the loss functions used for the generator and discriminator are shown in Equations (5) and (6) below:(5)$$L_{MLM\;}\left(x,\theta_{G}\right)=E\left(\;\begin{array}{c} \sum_{i\in m}^{n}-logp_{G}\left(x_{i}|x^{masked}\right) \end{array}\right)$$
(6)$$L_{Disc\;}\left(x,\theta_{D}\right)=E\left(\begin{array}{c} \sum_{t=1}^{n}-1\left(x_{t}^{corrupt}=x_{t}\right)\log D{(x}_{t}^{corrupt},t)-1(x_{t}^{corrupt}\;\neq x_{t})\;log(1-D{(x}_{t}^{corrupt},t)) \end{array}\right)$$

The minimized combined loss for both the generator and discriminator was given as
(7)$${\begin{array}{c} min\\ \theta_{G},\theta_{D} \end{array}}_{\;}\sum_{x\in X}L_{MLM\;}\left(x,\theta_{G}\right)+\lambda L_{Disc}\left(x,\theta_{D}\right)$$

### 4.5. Pretraining

As shown in Figure 5, the pretraining consisted of the generator and discriminator, which are essentially two transformer models. Here, the Generator corrupted a percentage of the tokens (i.e., amino acid residues) from the input fragments, and the discriminator was trained to detect the replaced tokens. This enabled the model to learn context-dependent biological properties of protein sequence fragments from a large-scale task-independent and unlabeled protein dataset. The patterns learned during this stage were then embedded into a smaller task-specific and labeled dataset in the downstream tasks, i.e., binary classification prediction for various protein–ion binding sites. This significantly reduced the amount of labeled data needed since the pretrained model had already learned the underlying patterns related to classification. We selected the ELECTRA-small model, which comprised 12 layers, 256 hidden layers, and 128-dimension embedding.

This model was chosen due to the relatively small size of our pretraining corpus and the fact that a larger-size model would have been computationally expensive to train, which may not have led to any significant improvement. The vocabulary size used was 25, which included all 20 amino acid residues, the ‘-’ character to pad positions at the protein terminus, [MASK] as the masking character, [CLS] to mark the start of a fragment, [SEP] to mark the end of a fragment, and [UNK] for out-of-vocabulary words, i.e., unknown amino acid residues. We masked 15% of each input fragment in the embedding layer, which was then encoded into the token embeddings matrix, having a dimension of [27 × 128]. Both the token and position embeddings were summed and presented as input tokens, i.e., *x* = [*x*_1_, …, *x*_27_], into the generator. The generator used was 25% of the size of the discriminator [32], with 12 layers and a hidden size of 64. The generator trained using maximum likelihood to predict the original masked-out amino acid residues based on the contextual information from neighboring amino acid residues in the protein fragment. The model was trained over 1 million training steps, using a batch size of 128 and a learning rate of 0.0001.

### 4.6. Fine-Tuning

After pretraining, the generator was discarded, and the discriminator was then fine-tuned using labeled data for various specific classification tasks. For this, a fully connected layer was built over the pretrained ELECTRA model and the entire network was fine-tuned with 12 layers of the discriminator. This was performed to ensure the error was backpropagated throughout the whole architecture and that the weights of the discriminator were updated based on the fragments in the fine-tuned dataset. We fine-tuned separate models for each ligand–ion binding site using labeled fragments generated from the protein sequence, as described in Table 1. The candidate binding residues used for the metals were C, H, E, and D, while the ones used for acidic radicals were G, H, K, R, and S. The training, testing and dev fragments were split by a ratio of 80%, 10%, and 10%, respectively. We added a fully connected layer at the end of the pretrained ELECTRA model and fine-tuned the entire network consisting of 12 layers of the discriminator, so that the error was backpropagated across the entire architecture and the discriminator weights were updated using the labeled data, as shown in Figure 6. Similar hyperparameters used in the pretraining were implemented at this stage, except for the learning rate and the number of training steps, which were set at 0.00001 and 200 epochs, respectively. Fine-tuning runs much quicker than pretraining.

### 4.7. Model Assessment

We evaluated IonPred using the following metrics: *Recall*, *Precision*, *F*1 score, Matthew’s correlation coefficient (*MCC*), and the Receiver operating characteristic (ROC) curve, which are defined below:(8)$$Recall=\frac{TP}{TP+FN}\times 100$$
(9)$$Precision=\frac{TP}{TP+FP}\times 100$$
(10)$$F1=2\times \frac{Precision\times Recall}{Precision+Recall}\times 100$$
(11)$$MCC=\frac{\left(TP\times TN\right)-\left(FP\times FN\right)}{\sqrt{\left(TP+FP\right)\times \left(TP+FN\right)\times \left(TN+FP\right)\times \left(TN+FN\right)}}$$

The ROC curve is a graphical representation used in binary classification to assess the performance of a model across all possible classification thresholds. It is used to understand the trade-off between the true positive rate (*TPR*) and false positive rate (1-specificity) at different threshold settings.
(12)$$TPR=\frac{TP}{TP+FN}$$
(13)$$FPR=\frac{FP}{FP+TN}\times 100$$
where *TP* represents the number of binding residues correctly predicted as binding residues, *TN* is the number of non-binding residues that are correctly predicted as non-binding residues, *FP* is the number of non-binding residues that are incorrectly predicted as binding residues, and *FN* represents the number of binding residues incorrectly predicted as non-binding residues.

We also reported the AUC score and AUPR score. These results are reported in Table 1 and Table 2.

## Figures and Tables

**Figure 1 molecules-28-06793-f001:**
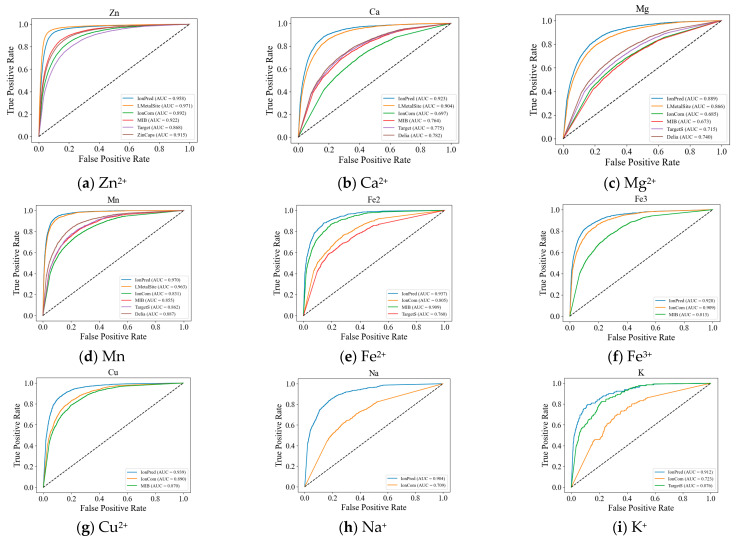
Comparison of ROC curves of IonPred with existing tools for the metal-ion predictions.

**Figure 2 molecules-28-06793-f002:**
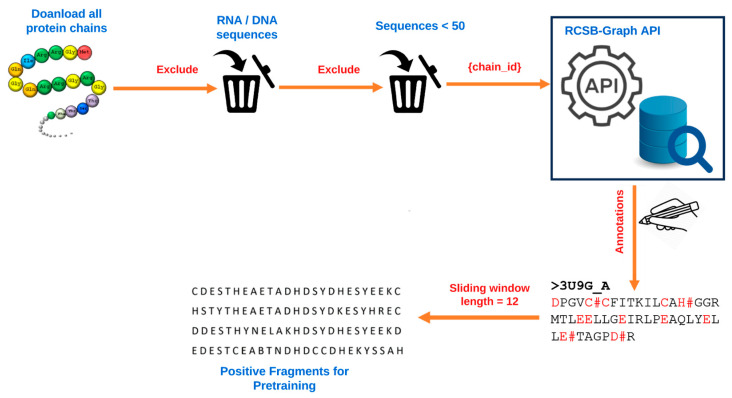
Data preprocessing for generating positive protein fragments used for pretraining.

**Figure 3 molecules-28-06793-f003:**
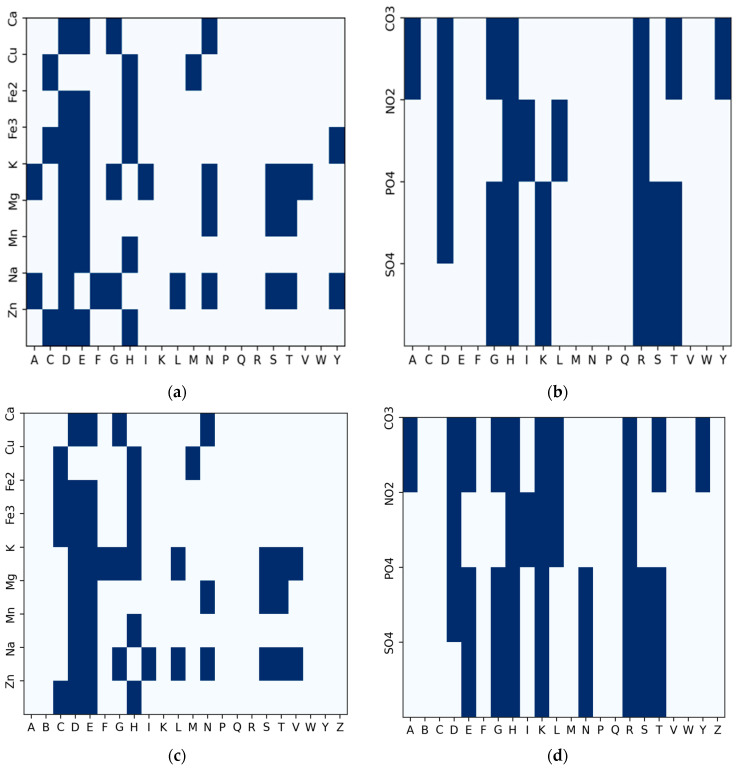
Frequency distribution of amino acid residues with respect to ion ligands derived from the IonCom dataset. (**a**) Metal ions (IonCom). (**b**) Acid radicals (IonCom). (**c**) Metal ions (BioLip). (**d**) Acid radicals (BioLip).

**Figure 4 molecules-28-06793-f004:**
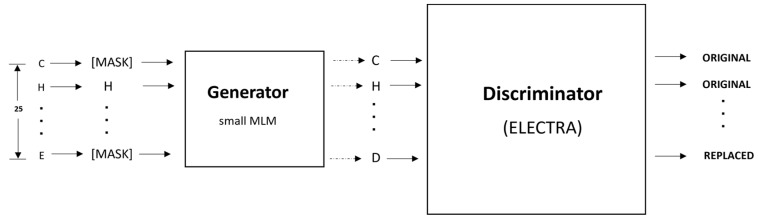
Electra architecture, which illustrates token corruption and replacement by the generator and discriminator.

**Figure 5 molecules-28-06793-f005:**
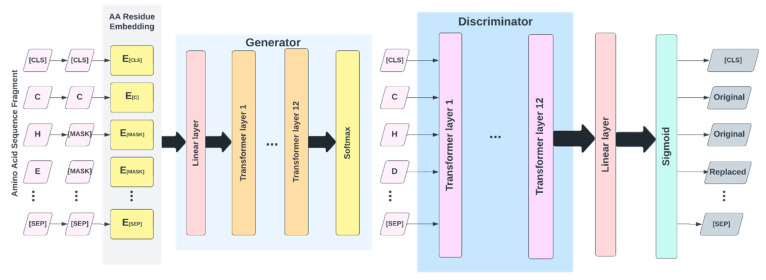
Pretraining process taking in raw protein fragments as input.

**Figure 6 molecules-28-06793-f006:**
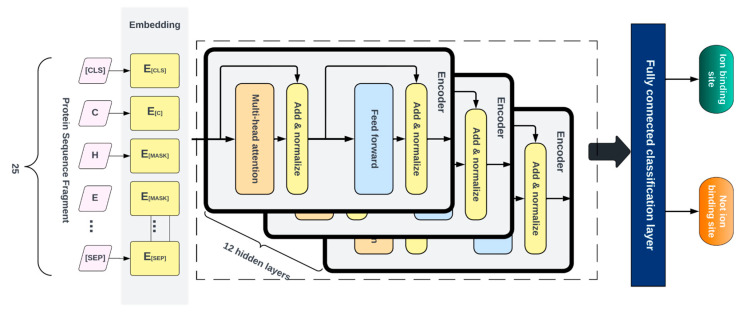
Fine-tuning with labeled dataset to generate probabilities for binary classification.

**Table 1 molecules-28-06793-t001:** Performance comparison of IonPred with other tools on metal-ion test sets.

Ion	Method	Rec	Pre	*F*1	MCC	AUC	AUPR
	MIB	0.739	0.220	0.339	0.389	0.922	0.388
	TargetS	0.450	0.750	0.563	0.578	0.868	0.594
Zn^2+^	ZinCaps	0.753	0.780	0.766	0.601	0.915	0.768
	IonCom	0.779	0.137	0.233	0.317	0.892	0.671
	LMetalSite	0.722	**0.859**	0.785	**0.760**	**0.971**	**0.801**
	IonPred	**0.790**	0.840	**0.814**	0.600	0.958	0.780
	MIB	0.341	0.082	0.132	0.139	0.764	0.105
	TargetS	0.119	0.487	0.191	0.244	0.775	0.165
Ca^2+^	DELIA	0.172	0.630	0.270	0.330	0.782	0.251
	IonCom	0.297	0.247	0.270	0.258	0.697	0.166
	LMetalSite	0.413	0.720	0.525	0.540	0.904	0.490
	IonPred	**0.467**	**0.759**	**0.578**	**0.615**	**0.923**	**0.520**
	MIB	0.246	0.043	0.073	0.082	0.673	0.053
	TargetS	0.118	0.491	0.190	0.237	0.715	0.148
Mg^2+^	IonCom	0.240	0.250	0.245	0.237	0.685	0.184
	DELIA	0.129	0.065	0.086	0.287	0.740	0.198
	LMetalSite	0.245	0.728	0.367	0.419	0.866	0.316
	IonPred	**0.400**	**0.780**	**0.529**	**0.470**	**0.889**	**0.450**
	MIB	0.462	0.096	0.159	0.193	0.855	0.168
	TargetS	0.271	0.496	0.350	0.362	0.862	0.322
Mn^2+^	DELIA	0.502	0.665	0.572	0.574	0.887	0.489
	IonCom	0.511	0.245	0.331	0.344	0.831	0.304
	LMetalSite	0.613	**0.719**	**0.662**	0.661	0.963	0.625
	IonPred	**0.620**	0.700	0.658	**0.670**	**0.970**	**0.670**
	MIB	0.586	0.620	0.603	0.573	0.909	0.354
Fe^2+^	TargetS	0.345	0.254	0.293	0.245	0.760	0.299
	IonPred	**0.749**	**0.728**	**0.738**	**0.723**	**0.937**	**0.771**
	IonCom	0.610	0.498	0.548	0.579	0.909	0.567
Fe^3+^	MIB	0.474	0.399	0.433	0.383	0.813	0.438
	IonPred	**0.743**	**0.612**	**0.671**	**0.652**	**0.928**	**0.724**
	IonCom	0.596	0.398	0.477	0.592	0.890	0.399
Cu^2+^	MIB	0.466	0.280	0.350	0.358	0.870	0.419
	IonPred	**0.789**	**0.634**	**0.703**	**0.620**	**0.939**	**0.677**
	IonCom	0.210	0.178	0.193	0.160	0.723	0.156
K^+^	TargetS	0.389	0.411	0.400	0.341	0.876	0.336
	IonPred	**0.498**	**0.672**	**0.572**	**0.524**	**0.912**	**0.478**
Na^+^	IonCom	0.451	0.292	0.355	0.218	0.709	0.233
	IonPred	**0.523**	**0.731**	**0.610**	**0.595**	**0.904**	**0.487**

Rec refers to Recall, Pre refers to Precision, MCC refers to Matthew’s correlation coefficient, *F*1 refers to *F*1 score, AUC refers to Area under the curve, and AUPR refers to Area under precision recall curve. Bold font indicates metric with best performance.

**Table 2 molecules-28-06793-t002:** Performance comparison of IonPred with other tools on non-metal-ion test sets.

Radicals	Method	Rec	Pre	*F*1	MCC	AUC	AUPR
CO_3_^2−^	IonCom	0.610	0.498	0.548	0.579	0.909	0.567
	IonPred	**0.743**	**0.612**	**0.671**	**0.652**	**0.928**	**0.724**
NO_2_^−^	IonCom	0.596	0.398	0.477	0.592	0.890	0.399
	IonPred	**0.789**	**0.634**	**0.703**	**0.620**	**0.939**	**0.677**
SO_4_^3−^	IonCom	0.210	0.178	0.193	0.160	0.723	0.156
	IonPred	**0.389**	**0.411**	**0.400**	**0.341**	**0.876**	**0.336**
PO_4_^3−^	IonCom	0.451	0.292	0.355	0.218	0.709	0.233
	IonPred	**0.523**	**0.731**	**0.610**	**0.595**	**0.904**	**0.487**

Rec refers to Recall, Pre refers to Precision, MCC refers to Matthew’s correlation coefficient, *F*1 refers to *F*1 score, AUC refers to Area under the curve, and AUPR refers to Area under precision recall curve. Bold font indicates metric with best performance.

**Table 3 molecules-28-06793-t003:** Performance evaluation of several ELECTRA model configurations on Zinc dataset.

Configuration	AUC	AUPR
ELECTRA-0.25G-100K	0.916	0.698
ELECTRA-0.25G-200K	0.951	0.756
IonPred-0.25G-1M	**0.958**	**0.780**
ELECTRA-0.5G-200K	0.926	0.739
ELECTRA-1.0G-200K	0.904	0.676
ELECTRA-no-pretraining	0.857	0.519

Bold font indicates metric with best performance.

**Table 4 molecules-28-06793-t004:** Sample predictions of known proteins that bind to Fe^3+^ and Mg^2+^.

Protein	Residue	Residue Position	Predicted Probability
3GKR_A (Fe^3+^)	D	65	0.558
	D	67	0.551
	D	151	0.516
	E	309	0.410
	E	311	0.499
3DHG_D (Mg^2+^)	E	104	0.891
	E	134	0.912
	H	137	0.896
	E	197	0.903
	E	231	0.920
	H	234	0.899

**Table 5 molecules-28-06793-t005:** Statistics of the residue distribution of each ion dataset used for fine-tuning.

Category	Ion	N_prot_	R_pos_	R_neg_
Metal ions	Ca^2+^	179	1360	119,192
	Cu^2+^	110	535	38,488
	Fe^2+^	227	1115	73,813
	Fe^3+^	103	439	34,113
	K^+^	53	536	18,776
	Mg^2+^	103	391	76,382
	Mn^2+^	379	1778	148,618
	Na^+^	78	489	27,408
	Zn^2+^	142	697	93,952
Acid radicals	CO_3_^2−^	62	316	22,766
	NO_2_^−^	22	98	8144
	PO_4_^3−^	303	2125	99,729
	SO_4_^2−^	339	2168	112,279

N_prot_ represents the number of protein chains, while R_pos and_ R_neg_ represent the number of binding residues and the number of non-binding residues, respectively.

**Table 6 molecules-28-06793-t006:** Statistics of the training, test, and validation fragments used for fine-tuning.

Category	Ion	Training	Test	Validation
Metal ions	Ca^2+^	849,087	108,857	95,919
	Cu^2+^	23,977	4074	3070
	Fe^2+^	51,398	6589	6345
	Fe^3+^	106,114	13,604	13,100
	K^+^	26,864	5848	6010
	Mg^2+^	594,193	76,179	73,357
	Mn^2+^	195,499	24,065	24,672
	Na^+^	46,070	7450	5493
	Zn^2+^	712,169	104,856	91,922
Acid radicals	CO_3_^2−^	11,465	1919	1417
	NO_2_^−^	9057	1305	1180
	PO_4_^3−^	114,234	23,836	13,240
	SO_4_^2−^	76,134	12,937	11,534

## Data Availability

All the source codes and data used for this project are available at https://github.com/clemEssien/IonPred (accessed on 12 September 2023).

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
