# Peer review of "Prediction of Protein Ion–Ligand Binding Sites with ELECTRA"

_molecules, 2023, doi:10.3390/molecules28196793_

Round 1

Reviewer 1 Report

The authors propose to use IonPred, a sequence-based approach that employs ELECTRA (Efficiently Learning an En-coder that Classifies Token Replacements Accurately) to predict ion binding sites in proteins on the base of raw protein sequences. They fine-tuned their pretrained model to predict the binding sites for nine metal ions (Zn2+, Cu2+, Fe2+, Fe3+, Ca2+, Mg2+, Mn2+, Na+, and K+) and four acid radical ion ligands (CO32−, SO42−, PO43−, NO2). They showed that their method is more computationally efficient than existing tools. The work certainly deserves publication but I have some comments:

Table 1. It would be good to explain the meaning of abbreviations Rec, Pre, F1, MCC, AUC and AUPR in the legend of the Table.

It would be helpful to explain the meaning of ROC curves in more detail.

It would be good to give some specific examples of the work of the proposed method using proteins with well-established binding sites for metal cations.

Reviewer 2 Report

This manuscript proposes IonPred, a new transformer model for ion binding site prediction. There are several major concerns.

1. The treatment of the BioLiP dataset splitting is unclear. According to line 224 to 230, BioLiP is 30% non-redundant (which I am not sure is true, as there is no such download option). If it is 30% non-redundant, how the dataset can be split by cd-hit using a 40% identity cutoff? Moreover, how large is the training/validation/testing set?

2. There are numerous mentioning of a fragment length of 25. Why this specific length? Is this the length limitation of the ELECTRA model? In other words, does that mean for a query protein, IonPred needs to first obtain fragments by sliding window, and then obtain the prediction on each fragment? This is probably not the best usage of Transformer, as all long-range interactions are ignored even though Transformer is known to handle long-range interactions.

3. The particular choice of pre-training set is not explained well. According to Figure 4, the generator and discriminator can be trained in a self-supervised manner before fine tuning. If this was true, there would be no need to limit the pre-training set only on RCSB; a larger and more comprehensive unlabeled set, e.g., UniRef50 would have been more appropriate.

4. Figure 3 was obtained on the RCSB pretraining set but the fine-tuning was on BioLiP, as the authors claimed that not all ions are biologically relevant. If this was true, why Figure 3 was not obtained based on BioLiP?

Round 2

Reviewer 1 Report

I am satisfied by the improvements of the text.

Author Response

Thank you for your support!

Reviewer 2 Report

The new revision claimed that " We used positive fragments for pretraining because through experimentation, we determined that pretraining with positive fragments made it easier to learn features related to ion-binding fragments more effectively." There is no data to support this claim.

I previously commented that "Figure 3 was obtained on the RCSB pretraining set but the fine-tuning was on BioLiP, as the authors claimed that not all ions are biologically relevant. If this was true, why Figure 3 was not obtained based on BioLiP?" The authors responded that "IonPred used a two-step training procedure. The first is pretraining which is to learn a comprehensive contextual representation of protein sequence segments for ion-binding sites. For this stage, we used labeled data. As we did pretraining with positive fragments, we generated Figure 3 based on the data obtained from RCSB. For the second stage (i.e., fine-tuning) uses a supervised learning method and labeled data to finetune the model. Any dataset can be used for this step." I still do not understand the response. If Figure 3 requires positive labels, it should use a dataset where the ion binding site's biologically relevance can be affirmed, such as BioLiP or the IonCom dataset. Why is RCSB used instead?
